# Mitigating Think-Answer Mismatch in LLM Reasoning Through Noise-Aware Advantage Reweighting

## Abstract

Group-Relative Policy Optimization (GRPO) is a key technique for training large reasoning models, yet it suffers from a potential challenge: the *Think-Answer Mismatch*, where noisy reward signals corrupt the learning process. This problem is most severe in unbalanced response groups, paradoxically degrading the signal precisely when it should be most informative. To address this challenge, we propose Stable Group-Relative Policy Optimization (S-GRPO), a principled enhancement that derives optimal, noise-aware advantage weights to stabilize training. Our comprehensive experiments on mathematical reasoning benchmarks demonstrate S-GRPO's effectiveness and robustness. On various models, S-GRPO significantly outperforms DR. GRPO, achieving performance gains of +2.5% on Qwen-Math-7B-Base, +2.2% on Llama-3.2-3B-Base, and +2.4% on Qwen-Math-1.5B-Instruct. Most critically, while standard GRPO fails to learn under 20% synthetic reward noise, S-GRPO maintains stable learning progress. These results highlight S-GRPO's potential for more robust and effective training of large-scale reasoning models. Our code is available at this anonymous repository.

## 1 Introduction

Recent breakthroughs in large-scale reasoning models are largely attributed to methods like Group-Relative Policy Optimization (GRPO) (Shao et al., 2024), a reinforcement learning technique that has propelled models such as DeepSeek-R1 and Qwen3 to state-of-the-art performance on challenging reasoning benchmarks (Guo et al., 2025; Shao et al., 2024; Liu et al., 2025). GRPO's efficacy stems from a simple yet powerful principle: rewarding responses that yield correct final answers relative to a group of sampled outputs, while penalizing incorrect ones. This approach obviates the need for an explicit value function, significantly simplifying the training pipeline.

Despite its empirical success, GRPO's reliance on final-answer correctness as a proxy for reasoning quality presents a potential challenge. A correct answer does not necessarily imply valid or logically sound reasoning (Tyen et al., 2023; Zheng et al., 2024; Song et al., 2025). For instance, Zheng et al. (2024) report that Qwen and LLaMA models exhibit reasoning error rates ranging from 3.5% to 51.8% across various benchmarks, even when their final answers are correct. Conversely, an incorrect final answer does not always indicate flawed reasoning. As shown by Kiciman et al. (2023), models like GPT-3.5 can produce intermediate reasoning steps that successfully identify and correct earlier mistakes, yet ultimately revert to an incorrect final answer. This phenomenon, commonly referred to as the *Think-Answer Mismatch* (Yao et al., 2025; Chen et al., 2025b), has been consistently observed across diverse models, tasks, and evaluation protocols.

The consequences of this misalignment are particularly pronounced for GRPO. Our analysis uncovers a specific failure mode: vulnerability to reward noise in unbalanced groups. As illustrated in Figure 1, a single false positive mismatch, where a response with flawed reasoning that happens to yield the correct answer, can have a disproportionately large impact depending on group composition. In a highly unbalanced group (e.g., one correct answer out of eight), this single mismatch sample can severely distort the advantage signal, inflating the overall observed advantage by up to 60% (5.31 vs. 3.32) compared to a balanced group. This creates a paradox: precisely when the learning signal should be strongest, that is, when a rare success occurs among many failures, it be-

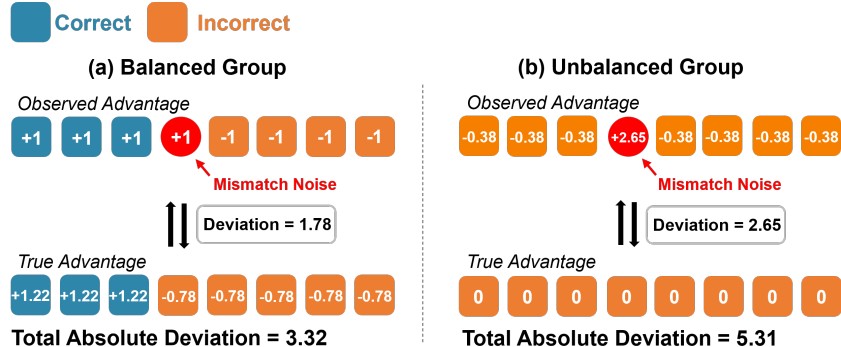

Figure 1: The impact of a single Think-Answer Mismatch on GRPO's advantage calculation in a balanced group (4 correct, 4 incorrect responses) versus an unbalanced group (1 correct, 7 incorrect responses). A false positive, where flawed reasoning leads to a correct answer (indicated by the red circle), causes a much larger deviation in the unbalanced group.

comes most vulnerable to corruption. As demonstrated in Section 2, this vulnerability is not merely theoretical; under noise levels (e.g., 20%), the standard GRPO learning process can collapse entirely.

To address this potential challenge, we propose Stable Group-Relative Policy Optimization (S-GRPO), a principled enhancement that explicitly models and mitigates the impact of reward noise. Our approach is founded on the insight that balanced groups provide an inherently more robust training signal. S-GRPO operationalizes this by deriving an optimal, closed-form advantage weight that minimizes the expected squared error between the observed and true advantages under a symmetric noise model. This reweighting scheme automatically down-weights signals from unbalanced groups where noise has an outsized impact, yielding a method that maintains the computational efficiency of GRPO while gracefully handling noisy rewards.

Our comprehensive evaluations demonstrate that S-GRPO consistently outperforms standard GRPO under identical experimental setups, achieving significant gains on models like Qwen-Math-7B (+2.5%), Llama-3.2-3B (+2.2%), and Qwen-Math-1.5B (+2.4%). More importantly, S-GRPO demonstrates remarkable robustness: while standard GRPO fails to learn under 20% synthetic label noise, S-GRPO maintains stable training progress with minimal performance degradation. Further analysis reveals that S-GRPO fosters a more stable and efficient training process, evidenced by smoother entropy reduction and the emergence of more reliable reasoning patterns.

Our contributions are three-fold:

- We are the first to identify and formalize the vulnerability of GRPO to the *Think-Answer Mismatch*, showing how its impact is amplified by group imbalance.
- We propose S-GRPO, a principled extension that derives optimal, noise-aware advantage weights to ensure robust policy updates.
- We demonstrate empirically that S-GRPO improves performance, robustness, and training stability across multiple reasoning benchmarks and model scales.

## 2 ROBUSTNESS OF GRPO TO REWARD NOISE

### 2.1 BACKGROUND: GROUP-RELATIVE POLICY OPTIMIZATION

Group-Relative Policy Optimization (GRPO) is a memory-efficient reinforcement learning algorithm designed for fine-tuning Large Language Models (LLMs). Its core innovation lies in eliminating the need for a separate critic model, a staple in traditional Reinforcement Learning from Human Feedback (RLHF) methods like Proximal Policy Optimization (PPO)(Schulman et al., 2017). Instead, GRPO computes the advantage for each response relative to a baseline derived from a group of peer responses generated for the same query, significantly reducing computational overhead during training.

For a given input query $q$, the actor LLM generates a group of $N$ responses, $\{o_i\}_{i=1}^{N}$. Each response is assigned a binary reward $r_i \in \{0, 1\}$, indicating whether it yields the correct final answer. The original GRPO framework defines the advantage by standardizing this reward within the group:

$$a_i = \frac{r_i - \bar{r}}{\sqrt{\bar{r}(1 - \bar{r}) + \epsilon}}, \tag{1}$$

where $\bar{r} = \frac{1}{N} \sum_{i=1}^{N} r_i$ is the empirical mean reward of the group and $\epsilon$ is a small constant to prevent division by zero. This group-wise normalization centers the advantages around zero and scales them to unit variance, which helps stabilize the learning process.

## 2.2 THE IMPACT OF THINK-ANSWER MISMATCH ON ADVANTAGE CALCULATION

We now analyze how a 'Think-Answer Mismatch' false positive, where reasoning is flawed but the final answer is correct, impacts advantage computation in GRPO. The impact of false negatives remains the same. Consider a group of $N$ responses where $k$ responses are observed to have a reward of 1. Under the original GRPO formulation, the observed positive advantage ($a_{\text{pos}}$) and negative advantage ($a_{\text{neg}}$) are:

$$a_{\text{pos}} = \frac{N - k}{\sqrt{k(N - k)}}$$
$$a_{\text{neg}} = \frac{-k}{\sqrt{k(N - k)}} \tag{2}$$

Now consider the case where one of these observed positive rewards is actually incorrect due to a Think-Answer Mismatch. The true reward distribution should have $(k-1)$ positive and $(N-k+1)$ negative responses. The corrected advantages become:

$$a_{\text{pos}}^{\text{true}} = \frac{N - k + 1}{\sqrt{(k - 1)(N - k + 1)}}$$
$$a_{\text{neg}}^{\text{true}} = \frac{-(k - 1)}{\sqrt{(k - 1)(N - k + 1)}} \tag{3}$$

The total absolute deviation in advantages across the entire group can be formulated as:

$$\Delta_{\text{total}} = \underbrace{\left| a_{\text{pos}} - a_{\text{neg}}^{\text{true}} \right|}_{\text{Mismatch Sample}}$$
$$+ \underbrace{(k - 1) \times \left| a_{\text{pos}} - a_{\text{pos}}^{\text{true}} \right|}_{\text{True Positives}}$$
$$+ \underbrace{(N - k) \times \left| a_{\text{neg}} - a_{\text{neg}}^{\text{true}} \right|}_{\text{True Negatives}} \tag{4}$$

This formulation reveals that a single Think-Answer Mismatch creates a cascading error: not only does the mismatch sample receive an incorrect advantage signal, but all other samples in the group also experience advantage distortions due to the altered group statistics. Figure 2 illustrates this effect for a group size of $N = 16$, which is used here for clarity, while all other experiments use $N = 8$. The deviation exhibits a U-shaped relationship with group composition, being most pronounced in imbalanced groups.

## 2.3 CONSEQUENCES FOR TRAINING DYNAMICS

To study how Think–Answer Mismatches affect training, we inject synthetic noise into the reward signal by randomly flipping the binary reward of a response. This simulates an increased rate of mismatches. We consider noise rates of 2%, 5%, 10%, and 20%. Figure 3 shows the training curves: each panel corresponds to one noise level and compares GRPO with S-GRPO.

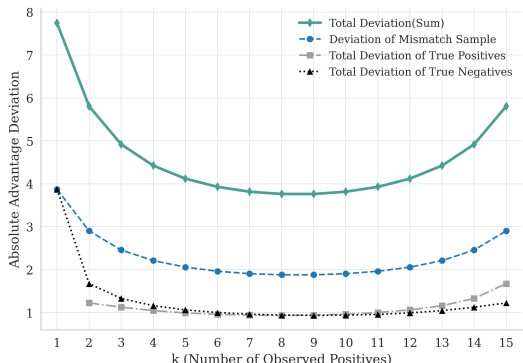

Figure 2: The impact of a single false positive Think-Answer Mismatch on GRPO's advantage calculation in groups of size $N = 16$.

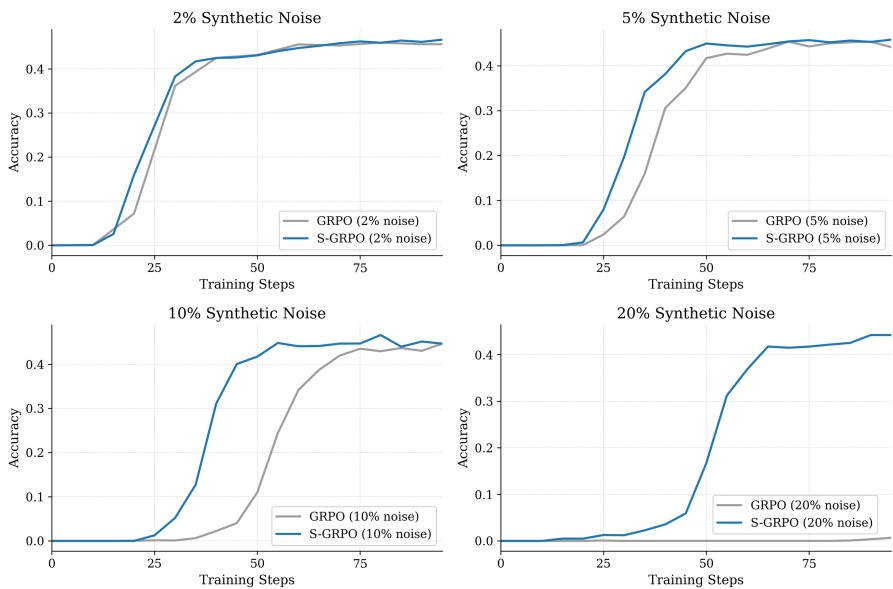

Figure 3: Impact of synthetic reward noise on the training dynamics of S-GRPO and standard GRPO. Pass@1 accuracy over 100 training steps for GRPO and S-GRPO under synthetic noise levels of 2%, 5%, 10% and 20%.

Under mild injected noise levels (2%, 5%, and 10%), both standard GRPO and our proposed S-GRPO learn effectively, with S-GRPO consistently reaching a slightly higher final accuracy. When the noise level increases to 20%, however, GRPO's performance almost collapses, showing little meaningful improvement within the first 100 steps. In contrast, S-GRPO continues to learn effectively under all noise levels. These results demonstrate that as the rate of Think–Answer Mismatches increases, standard GRPO's performance degrades substantially, while S-GRPO maintains robust learning capabilities.

# 3 S-GRPO: STABLE GRPO THROUGH NOISE-AWARE REWEIGHTING

To mitigate the adverse effects of Think-Answer Mismatches, we introduce S-GRPO (Stable GRPO), which denoises the advantage signal through principled reweighting based on an explicit noise model.

## 3.1 SYMMETRIC REWARD NOISE MODEL

We model the Think-Answer Mismatch as symmetric label noise Angluin & Laird (1988); Van Rooyen et al. (2015). Each observed reward $r_i$ is an independent flip of the latent true reward $r_i^* \in \{0, 1\}$ with a fixed probability $p$:

$$\mathbb{P}(r_i \neq r_i^*) = p, \quad \text{where } 0 \leq p < 0.5. \tag{5}$$

Here, $p$ is a hyperparameter representing the probability of a Think-Answer Mismatch. In practice, this value tends to be higher for more complex datasets and weaker models, and lower for simpler datasets and stronger models.

Given the observed mean reward $\bar{r} = k/N$ for a group of $N$ responses, the expected true mean reward $t = \mathbb{E}[r_i^*]$ can be estimated as:

$$t = \frac{\bar{r} - p}{1 - 2p}. \tag{6}$$

This relationship follows from $\bar{r} = \mathbb{P}(r_i = 1) = (1-p)t + p(1-t)$. We clip $t$ to $[0, 1]$ to ensure validity. Intuitively, if the observed success rate $\bar{r}$ approaches the noise rate $p$, the estimated true success rate $t$ approaches 0.

## 3.2 DENOISED ADVANTAGE VIA OPTIMAL REWEIGHTING

Our goal is to find an optimal weight $w^\star$ for each group that minimizes the expected squared error between the reweighted observed advantage and the unobserved true advantage. The standardized advantages are:

$$a_i = \frac{r_i - \bar{r}}{\sigma_r} \quad \text{and} \quad a_i^* = \frac{r_i^* - t}{\sigma_t}, \tag{7}$$

where $\sigma_r^2 = \bar{r}(1 - \bar{r}) + \epsilon$ and $\sigma_t^2 = t(1 - t) + \epsilon$.

We seek to solve:

$$w^\star = \arg\min_w \mathcal{L}(w) = \mathbb{E}\left[(wa_i - a_i^*)^2\right]. \tag{8}$$

Since both $a_i$ and $a_i^*$ are standardized with zero mean and unit variance, expanding the loss function yields:

$$\mathcal{L}(w) = w^2 - 2w \operatorname{Cov}(a_i, a_i^*) + 1. \tag{9}$$

Setting the derivative to zero gives the optimal weight:

$$w^\star = \operatorname{Cov}(a_i, a_i^*) = \frac{\operatorname{Cov}(r_i, r_i^*)}{\sigma_r \sigma_t}. \tag{10}$$

The covariance between observed and true rewards is $\operatorname{Cov}(r_i, r_i^*) = (1 - 2p)t(1 - t)$. Substituting yields:

$$w^\star(N, k, p) = \frac{(1 - 2p)t(1 - t)}{\sqrt{\bar{r}(1 - \bar{r}) + \epsilon}\sqrt{t(1 - t) + \epsilon}}. \tag{11}$$

This weight represents the correlation coefficient between observed and true rewards, scaled by the noise factor $(1 - 2p)$. S-GRPO uses the reweighted advantage $w^\star a_i$ for policy gradient updates. Since correlation is invariant under positive affine rescaling of the reward values, the same formula applies to alternative binary encodings such as $\{-1, 1\}$ rewards (see Appendix F.1 for details). For more general continuous rewards, Eq. (8) still yields $w^\star = \operatorname{Cov}(r, r^*)/(\sigma_r \sigma_{r^*})$, and specializing this expression to a particular noise model is left to future work (Appendix F.2).

### 3.2.1 ANALYSIS OF THE OPTIMAL WEIGHT

The behavior of $w^\star$ (Equation 11) reveals several key properties aligned with our goal of robust learning:

As illustrated in Figure 4, the weight $w^\star$ exhibits three important characteristics:

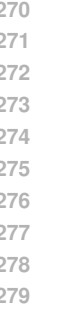
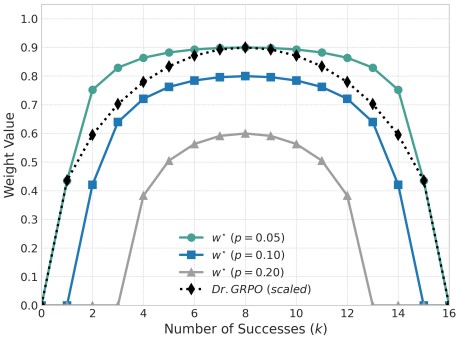

Figure 4: The optimal weight $w$ as a function of successful responses $k$ in a group of size $N = 16$ for different assumption noise levels $p$. Dr. GRPO's reweighting strategy (scaled to maximum 0.9) is shown for comparison.

1. **Noise-Adaptive Attenuation:** The weight is bounded by $(1-2p)$, with higher noise levels uniformly down-weighting the learning signal. In the noiseless case ($p = 0$), $w^\star = 1$, recovering the original GRPO.

2. **Confidence through Consensus:** The weight is smallest for highly imbalanced groups and largest for balanced ones ($k \approx N/2$). This concave shape formalizes the intuition that balanced groups provide more reliable signals.

3. **Noise-Gating Mechanism:** When the observed success rate falls below the assumed noise rate $p$, the weight becomes zero. For example, with $p = 0.20$, groups with $k \leq 3$ or $k \geq 13$ (out of 16) are completely gated, preventing updates based on statistically unreliable signals.

**Comparison with Existing Methods.** Figure 4 also shows Dr. GRPO's heuristic (Liu et al., 2025), which removes standard deviation normalization. While both approaches upweight balanced groups, Dr. GRPO lacks noise adaptivity and the hard gating mechanism for low-confidence cases, properties that prior heuristics like DAPO (Yu et al., 2025) and Seed-GRPO (Chen et al., 2025a) also cannot provide.

### 3.3 OPTIMIZATION OBJECTIVE

We adopt a clipped surrogate objective inspired by PPO, adapted for noise-aware reweighting:

$$\mathcal{L}_i(\theta) = \min \big(\text{ratio}_i(\theta) w^\star a_i,$$
$$\text{clip}(\text{ratio}_i(\theta), 1 - \epsilon, 1 + \epsilon) w^\star a_i\big), \tag{12}$$

where $\text{ratio}_i(\theta) = \frac{\pi_\theta(o_i|q)}{\pi_{\theta_{\text{old}}}(o_i|q)}$ is the importance sampling ratio.

The overall training objective for query $q$ with $G$ responses is:

$$\mathcal{L}(\theta) = \frac{1}{G} \sum_{i=1}^{G} \mathcal{L}_i(\theta). \tag{13}$$

Model parameters are updated via stochastic gradient ascent following the standard PPO scheme.

## 4 EXPERIMENTS

In this section, we empirically evaluate S-GRPO across multiple dimensions. We investigate the following research questions:

- **RQ1**: How does S-GRPO compare to baselines on mathematical reasoning benchmarks?

- **RQ2**: How do different noise assumptions (p values) affect S-GRPO's performance and training stability?

- **RQ3**: What emergent behaviors and characteristics does S-GRPO exhibit compared to standard GRPO?

### 4.1 EXPERIMENTAL SETUP

#### 4.1.1 DATASETS

We conduct reinforcement learning on 8,500 problems sampled from the MATH dataset (Hendrycks et al., 2021b), specifically selecting problems with difficulty levels 3-5 to ensure appropriate challenge for our models. For evaluation, we employ four standard mathematical reasoning benchmarks. AMC (83 problems), MATH500 (500 problems) , Minerva (272 problems)(Lewkowycz et al., 2022), OlympiadBench (475 problems)(Huang et al., 2024). We exclude the commonly used AIME24 benchmark due to its limited size (30 problems), which leads to unstable results, particularly for smaller models[1].

#### 4.1.2 BASELINES

We compare S-GRPO against several strong baselines representing the current state-of-the-art in mathematical reasoning. Among the advanced reasoning models, we include RAFT++ (Xiong et al., 2025), OpenReasoner-Zero-7B (Hu et al., 2025) and SimpleRL-Zoo-7B (Zeng et al., 2025).

For the most direct comparison to our approach, we evaluate against: the original GRPO (Shao et al., 2024), and Dr. GRPO (Liu et al., 2025). These two baselines are particularly important as they share nearly identical experimental settings with S-GRPO, providing the fairest comparison for evaluating our noise-aware reweighting strategy.

#### 4.1.3 EVALUATION METRICS

We report Pass@1 accuracy as our primary metric across all benchmarks, using greedy sampling for deterministic evaluation. To address the inherent instability in RL training, we report the average of the top-3 checkpoint performances, evaluated every 16 training steps within a maximum of 500 steps. This approach provides a more robust assessment of model capabilities while accounting for training variance.

#### 4.1.4 TRAINING DETAILS

We train S-GRPO on three diverse base models to ensure generalizability: Qwen2.5-Math-7B-Base, Qwen2.5-Math-1.5B-Instruct, and Llama-3.2-3B-Base. This selection covers different model scales from 1.5B to 7B parameters and includes both base and instruction-tuned variants, addressing concerns about method sensitivity to base model choice (Zuo et al., 2025; Shao et al., 2025). Additional hyperparameters and optimization details are provided in Appendix A.

### 4.2 MAIN RESULTS (RQ1)

Table 1 presents our main experimental results across all benchmarks and models. Comparison with State-of-the-Art Methods. S-GRPO demonstrates strong performance against competitive baselines. Our 7B model achieves an average accuracy of 56.0%, shows S-GRPO as a highly competitive approach for mathematical reasoning.

Since RL performance is highly sensitive to base models and hyperparameters, the most informative comparisons are with GRPO and Dr. GRPO under identical experimental settings. S-GRPO consistently outperforms both baselines across all three base models. On Qwen2.5-Math-1.5B-Instruct, S-GRPO achieves 49.7% average accuracy, representing a 2.4 percentage point improvement over Dr. GRPO. Similar gains are observed on Llama-3.2-3B-Base (+2.2 points) and Qwen2.5-Math-7B-Base (+2.5 points). The consistency of these improvements across diverse model architectures and

---

[1]See `https://github.com/sail-sg/understand-r1-zero/issues/21` for discussion on AIME24's instability.

Table 1: Performance comparison across mathematical reasoning benchmarks. Results marked with ⋆ indicate experiments we conducted under identical settings. Other results are from original papers. We report mean ± standard deviation for the top-3 checkpoints. The best results from models with same size are in bold.

| Model | AMC | MATH500 | Minerva | OlympiadBench | Average |
|---|---|---|---|---|---|
| *State-of-the-art reasoning models* | | | | | |
| RAFT++ 7B | - | 80.5 | 35.8 | 41.2 | - |
| OpenReasoner-Zero-7B | 47.0 | 79.2 | 31.6 | 44.0 | 50.5 |
| SimpleRL-Zoo-7B | 60.2 | 78.2 | 27.6 | 40.3 | 51.6 |
| *Qwen2.5-Math-1.5B-Instruct* | | | | | |
| Base Model | 43.4 | 61.8 | 15.1 | 28.4 | 37.2 |
| GRPO-1.5B⋆ | $47.0_{\pm 2.4}$ | $74.0_{\pm 0.4}$ | $23.2_{\pm 0.4}$ | $39.3_{\pm 0.6}$ | $46.3_{\pm 0.4}$ |
| Dr. GRPO-1.5B⋆ | $48.2_{\pm 2.4}$ | $75.8_{\pm 0.6}$ | $25.0_{\pm 0.7}$ | $40.1_{\pm 0.7}$ | $47.3_{\pm 0.5}$ |
| **S-GRPO-1.5B⋆** | $\mathbf{51.8}_{\pm 1.2}$ | $\mathbf{77.8}_{\pm 0.2}$ | $\mathbf{27.6}_{\pm 0.4}$ | $\mathbf{42.2}_{\pm 0.3}$ | $\mathbf{49.7}_{\pm 0.3}$ |
| *Llama-3.2-3B-Base* | | | | | |
| Base Model | 2.4 | 6.4 | 6.3 | 1.3 | 3.3 |
| GRPO-3B⋆ | $\mathbf{7.2}_{\pm 1.2}$ | $12.2_{\pm 0.8}$ | $10.3_{\pm 1.1}$ | $3.5_{\pm 0.6}$ | $8.3_{\pm 0.4}$ |
| Dr. GRPO-3B | **7.2** | 10.0 | 11.0 | 2.2 | 7.6 |
| **S-GRPO-3B⋆** | $\mathbf{7.2}_{\pm 1.2}$ | $\mathbf{14.2}_{\pm 0.4}$ | $\mathbf{12.9}_{\pm 0.7}$ | $\mathbf{4.8}_{\pm 0.4}$ | $\mathbf{9.8}_{\pm 0.4}$ |
| *Qwen2.5-Math-7B-Base* | | | | | |
| Base Model | 45.8 | 69.0 | 21.3 | 34.7 | 42.7 |
| GRPO-7B⋆ | $57.8_{\pm 3.6}$ | $79.2_{\pm 1.8}$ | $29.4_{\pm 1.5}$ | $41.5_{\pm 2.1}$ | $51.5_{\pm 1.3}$ |
| Dr. GRPO-7B | **62.7** | 80.0 | 30.1 | 41.0 | 53.5 |
| **S-GRPO-7B⋆** | $61.5_{\pm 2.4}$ | $\mathbf{82.2}_{\pm 1.4}$ | $\mathbf{35.7}_{\pm 1.1}$ | $\mathbf{45.3}_{\pm 1.5}$ | $\mathbf{56.0}_{\pm 0.4}$ |

scales validates our theoretical analysis: noise-aware reweighting provides a principled solution to the Think-Answer Mismatch problem.

### 4.3 ROBUSTNESS TO NOISY REWARDS (RQ2)

While Section 2.3 demonstrated S-GRPO's robustness under synthetic noise injection, here we examine how different noise assumption levels of $p$ affect training dynamics in practical settings.

#### 4.3.1 TRAINING DYNAMICS UNDER DIFFERENT NOISE ASSUMPTIONS

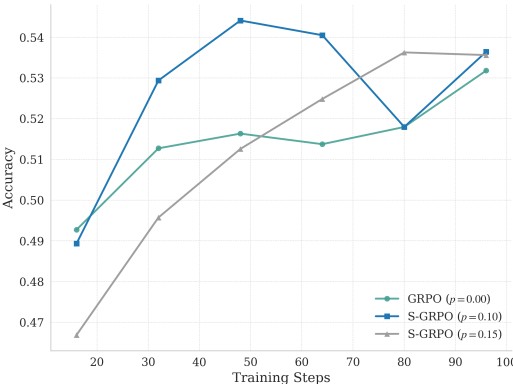

Figure 5: Training dynamics of S-GRPO under different noise assumptions $p$ (0, 0.10, 0.15) on Qwen2.5-Math-7B-Base during the first 100 training steps.

The results reveal a fundamental trade-off between learning speed and stability. With $p = 0.10$, the model achieves rapid initial improvement, jumping from 49% to 54% accuracy within 50 steps,

significantly outperforming baseline GRPO which plateaus around 51-52%. However, performance temporarily dips around step 80 before recovering. In contrast, $p = 0.15$ exhibits more conservative behavior: starting from a lower baseline (46%), it shows steady, monotonic improvement throughout training, ultimately converging to similar performance levels by step 100.

This behavior stems from S-GRPO's noise-gating mechanism. At $p = 0.15$, groups with extreme imbalance ($k \in \{1, 7\}$ for $N = 8$) receive zero weight, effectively filtering out potentially misleading signals. While this initially slows learning, it prevents corrupted updates from groups where a single mismatch could dominate the advantage calculation. The resulting trade-off suggests practitioners should choose $p$ based on their requirements: lower values for rapid initial gains, higher values for monotonic improvement and long-term stability.

### 4.3.2 POLICY ENTROPY EVOLUTION

Recent work has identified the balance between exploration (policy entropy) and exploitation (accuracy) as crucial for RL-based reasoning models (Cheng et al., 2025; Cui et al., 2025). Figure 6 tracks entropy evolution during training, revealing how noise-aware reweighting affects exploration-exploitation balance.

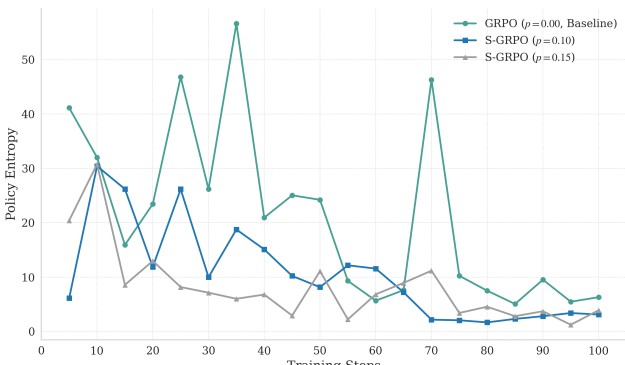

Figure 6: Policy entropy evolution under different noise assumptions on Qwen2.5-Math-7B-Base. Higher noise assumptions lead to smoother entropy reduction.

Without reweighting ($p = 0$), policy entropy exhibits severe instability with dramatic fluctuations. This erratic behavior indicates the model alternates between near-deterministic policies and complete uncertainty, suggesting that conflicting signals from mismatched data cause repeated abandonment of learned behaviors.

In contrast, S-GRPO demonstrates controlled entropy reduction. At both $p = 0.10$ and $p = 0.15$, entropy decreases smoothly with only minor fluctuations. This monotonic decrease indicates consistent exploration throughout training, with gradual transition to exploitation.

### 4.4 ANALYSIS OF S-GRPO CHARACTERISTICS (RQ3)

Beyond the core performance and robustness improvements demonstrated above, S-GRPO exhibits several distinctive characteristics compared to standard GRPO. These include enhanced self-reflection patterns, increased response detail, and improved reasoning coherence. Comprehensive analysis of these emergent behaviors, including ablation studies on optimal noise assumptions and qualitative case studies, is provided in Appendix C and Appendix D.

## 5 RELATED WORK

**GRPO and Variants.** Group-Relative Policy Optimization (GRPO) (Shao et al., 2024) revolutionized LLM reasoning training by eliminating value functions and computing advantages relative to peer responses. There are two major lines of algorithmic improvements. The first focuses on controlling exploration behavior for more effective learning, such as clipping higher to mitigate entropy

collapse (Yu et al., 2025) or selectively updating only the 20% of tokens with high entropy (Wang et al., 2025). These methods are orthogonal to ours and can be combined. The second line improves group-level reward computation, including Dr. GRPO (Liu et al., 2025), SEED-GRPO (Chen et al., 2025a), and Dang & Ngo (2025). However, these methods are primarily empirically motivated rather than grounded in noise theory. Our S-GRPO uniquely provides a principled, noise-aware reweighting solution derived from first principles.

**Think-Answer Mismatch.** The disconnect between correct answers and valid reasoning has been extensively documented (Lightman et al., 2023; Chen et al., 2025b), with error rates ranging from 3.5% to 51.8% even when final answers are correct. While process supervision methods (Luo et al., 2024; Lai et al., 2024) aims to address this through expensive step-level annotations, our approach handles the mismatch at the outcome level through principled noise modeling, maintaining computational efficiency while improving robustness.

**Noise-Robust Learning.** Learning from noisy rewards has been addressed through various approaches including robust MDPs (Iyengar, 2005), Bayesian methods (Yang et al., 2024), and symmetric loss functions (Nishimori et al., 2025). We adapt the classical symmetric noise model (Angluin & Laird, 1988) to derive closed-form optimal weights for GRPO. Unlike prior heuristic reweighting schemes, our approach is parameter-free given the noise estimate and requires no architectural changes or additional computation.

## 6 CONCLUSION

We identified and addressed a potential challenge in GRPO: its susceptibility to reward noise amplified by group imbalance. Our analysis revealed that a single mismatch sample can distort advantages by up to 60% in unbalanced groups than in balanced groups, causing standard GRPO to fail entirely under a high noise levels.

S-GRPO provides a principled solution through noise-aware optimal reweighting. The derived weight $w^\star$ elegantly implements three key properties: noise-adaptive attenuation, confidence through consensus, and automatic gating of unreliable signals. These emerge naturally from minimizing expected squared error rather than heuristic design.

Empirically, S-GRPO achieves consistent 2-3% improvements across diverse models while maintaining stable learning under 20% noise where standard GRPO collapses. Beyond accuracy gains, S-GRPO fundamentally improves training dynamics, including promoting smooth entropy reduction, increased self-reflection, and more detailed reasoning.

This work demonstrates that principled analysis of algorithmic vulnerabilities can yield simple yet effective solutions. As reasoning models become critical infrastructure, such robustness improvements are essential for reliable deployment. Future work could explore asymmetric noise models, integration with process supervision, and applications to other RL algorithms for LLMs.

ETHICS STATEMENT

This research adheres to the ICLR Code of Ethics. All data used in this study are publicly available and anonymized where necessary. No human subjects were directly involved. Potential biases in the datasets have been considered and mitigated. There are no conflicts of interest or sponsorship issues associated with this work.

REPRODUCIBILITY STATEMENT

We have made significant efforts to ensure the reproducibility of our work. All datasets used in our experiments are publicly available, and their preprocessing steps are described in Section 4.1.1. The implementation details of our models and training procedures are fully described in Sections 3 and 4.1.4. Additional hyperparameter settings and ablation studies are provided in Appendix A, while qualitative case analyses are included in Appendix B. Moreover, we release an anonymous repository containing the complete source code for our experiments: `https://anonymous.4open.science/r/S-GRPO-5F46`.

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

## A  EXPERIMENTAL SETUP

Our implementation is built upon the public Open-source Algorithm-driven Training (OAT) framework[2]. Our proposed S-GRPO method was implemented as a direct extension of the existing Dr. GRPO module within the OAT codebase. This approach ensures that our comparisons to baselines are fair, as we inherit the vast majority of the underlying training pipeline and focus specifically on the impact of our noise-aware reweighting strategy.

All experiments were conducted on a cluster of 8 NVIDIA A100 (80GB) GPUs, using BF16 mixed-precision for training. A typical 500-step training run for a 7B model required approximately 40 hours. The key hyperparameters for our experiments are summarized in Table 2.

Table 2: Key hyperparameters for S-GRPO training runs.

| Parameter | Value |
|---|---|
| *Core Algorithm Parameters* | |
| Base Framework | OAT |
| Learning Rate | $1 \times 10^{-6}$ (Constant scheduler) |
| S-GRPO Noise Level ($p$) | 0.10 (for 7B), 0.15 (for 1.5B/3B) |
| Rollouts per Prompt ($N$) | 8 |
| *Training & Generation Configuration* | |
| Global Training Batch Size | 128 |
| Global Rollout Batch Size | 128 |
| Max Generation Length | 4096 tokens |
| Temperature | 1.0 |
| Top-p | 0.95 |
| Evaluation Frequency | Every 16 steps |
| Base Models | Qwen2.5-Math-7B, Llama-3.2-3B, etc. |
| Training Data | 8,500 problems from MATH (levels 3-5) |
| Prompt Template | DeepSeek-R1 |

## B  CASE STUDY ANALYSIS

We provide detailed analysis of representative examples demonstrating how S-GRPO's mitigation of Think-Answer Mismatch during training leads to fundamentally different learned behaviors.

---

[2]https://github.com/sail-sg/oat

## B.1 Case 1: Coherence in Reasoning Framework

**Problem:** Let $p(x)$ be a polynomial of degree 5 such that $p(n) = \frac{n}{n^2-1}$ for $n = 2, \ldots, 7$. Find $p(8)$.

---

**Model trained with GRPO:**

To find the polynomial $p(x)$, we can use... However, a simpler approach is to use the polynomial:

$$p(x) = \frac{x}{(x^2 - 1)\frac{(x-2)(x-3)\ldots(x-7)}{(1-2)(1-3)\ldots(1-7)}}.$$

...The value of $p(8)$ is $\boxed{\dfrac{8}{63}}$.

- - - - - - - - - - - - - - - - - - - - - - - - - - - - - - - - - - - - - - - - -

**Model trained with S-GRPO:**

To solve for $p(8)$, **we can define a new polynomial** $q(x) = (x^2 - 1)p(x) - x$**.** This polyno-mial... has roots at $x = 2, 3, \ldots, 7$. Since $q(x)$ is a polynomial of degree 7, it can be written as:

$$q(x) = (x - 2)\ldots(x - 7)(x - r)$$

...The value of $p(8)$ is $\boxed{\dfrac{3}{56}}$.

---

**Analysis.** This example illustrates how Think–Answer Mismatch can corrupt GRPO training. The baseline model's derivation is mathematically incoherent: it mixes ideas from polynomial interpola-tion with unrelated algebraic manipulations. Such outputs arise when the training objective rewards the final answer regardless of the validity of the intermediate steps. During standard GRPO training, if this kind of reasoning occasionally yields the correct answer, the model can learn that abandoning logical coherence is an acceptable strategy.

S-GRPO's noise-aware reweighting mitigates this effect by reducing the influence of groups in which such mismatches are likely. In this example, the S-GRPO model instead constructs an auxil-iary polynomial $q(x)$ that closely resembles a standard textbook solution.

A natural concern is that penalizing shortcut-like solutions might push the model toward complex reasoning patterns that it has not yet mastered, increasing the risk of execution errors. In S-GRPO, this risk is limited by the way the group weights $w^\star(N, k, p)$ defined in Eq. (11) depend on the empirical success rate $k/N$. As discussed in Sec. 3.2.1, the weight is largest for balanced groups and becomes zero when the observed success rate falls below the assumed noise rate $p$. Groups whose apparent successes are unstable or indistinguishable from noise therefore receive very small weights and contribute little to the update.

Conversely, when a more structured strategy consistently improves the success rate across samples, its groups receive higher weights and are reinforced. As a result, S-GRPO tends to strengthen reasoning patterns that are both correct and reproducible, rather than complexity for its own sake. In this case study, the emergence of a coherent polynomial construction indicates that this more systematic approach is not only correct but also reliably reproducible across rollouts.

## B.2 Case 2: Fidelity in Computational Execution

**Problem:** Find the greatest integer less than $(\sqrt{7} + \sqrt{5})^6$.

**Model trained with GRPO:** ...we start by letting $x = \sqrt{7} + \sqrt{5}$ and $y = \sqrt{7} - \sqrt{5}$. Notice that $x^6 + y^6$ is an integer... **...By calculating the first few terms, we can see that**

$x^6 + y^6 = 2916$**.** ...the greatest integer less than $x^6$ is 2915. The answer is $\boxed{2915}$.

----------------------------------------

**Model trained with S-GRPO:** Let $a = \sqrt{7} + \sqrt{5}$ and $b = \sqrt{7} - \sqrt{5}$.

$$a^2 = 12 + 2\sqrt{35}$$
$$a^4 = (12 + 2\sqrt{35})^2$$
$$= 284 + 48\sqrt{35}$$
$$a^6 = a^2 \cdot a^4$$
$$= 6768 + 1144\sqrt{35}$$
$$\Rightarrow a^6 + b^6 = 13536$$

...the greatest integer less than $a^6$ is 13535. The answer is $\boxed{13535}$.

**Analysis:** This case reveals how Think-Answer Mismatch arises from insufficient context utilization when facing computational challenges. Both models correctly identify the conjugate approach, but their execution diverges dramatically.

The GRPO-trained model attempts to directly compute $x^6 + y^6$, which is a calculation requiring tracking binomial expansions with terms like $\binom{6}{k}(\sqrt{7})^k(\sqrt{5})^{6-k}$ that exceeds its reliable computational horizon. Unable to maintain context through this complex calculation, it resorts to hallucinating a value (2916 vs. correct 13536). This behavior emerges from training that rewards such computational overreach when lucky guesses yield correct final answers.

In contrast, the S-GRPO-trained model demonstrates learned respect for its computational boundaries through systematic decomposition. Each intermediate step ($a^2$, $a^4$, $a^6$) involves only a single algebraic operation within the model's capability, with explicit context propagation between steps. This incremental approach directly results from S-GRPO's training process: by down-weighting signals from unbalanced groups where computational overreach might accidentally succeed, it reinforces policies that favor verifiable, step-wise progress over high-risk leaps.

### B.3 QUALITATIVE ANALYSIS

We analyze how S-GRPO's mitigation of Think-Answer Mismatch during training produces fundamentally different learned behaviors compared to baseline GRPO.

On a polynomial interpolation problem ($p(n) = \frac{n}{n^2-1}$ for $n = 2, \ldots, 7$, find $p(8)$), the GRPO model outputs mathematical nonsense (a formula mixing unrelated concepts), which yields $\frac{8}{63}$. The S-GRPO model maintains coherent reasoning via auxiliary polynomial $q(x) = (x^2 - 1)p(x) - x$, though arriving at incorrect $\frac{5767}{63}$. This reveals how standard GRPO can learn to abandon logic when such gibberish accidentally succeeds during training.

For $(\sqrt{7}+\sqrt{5})^6$, GRPO attempts direct computation of $x^6+y^6$ beyond its capability and hallucinates "2916" (correct: 13536). S-GRPO instead decomposes systematically: $a^2 = 12 + 2\sqrt{35} \rightarrow a^4 = 284 + 48\sqrt{35} \rightarrow a^6 = 6768 + 1144\sqrt{35}$, maintaining context throughout.

These patterns demonstrate that S-GRPO's noise-aware reweighting during training filters out rewards from computational overreach and logical inconsistency, producing models that maintain reasoning coherence and respect computational boundaries.

# C ANALYSIS OF S-GRPO CHARACTERISTICS

## C.1 EMERGENCE OF SELF-REFLECTION PATTERNS

Self-reflection is considered a hallmark of effective reasoning (Guo et al., 2025; Min et al., 2024; Muennighoff et al., 2025). Following Liu et al. (2025), we analyze the frequency of self-reflection keywords in model outputs (Figure 7).

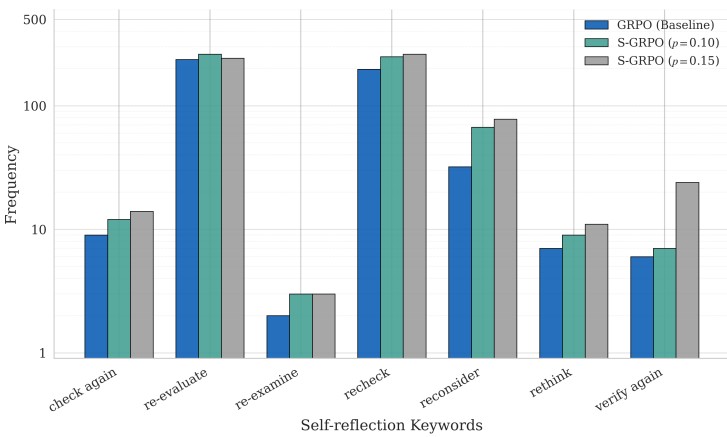

Figure 7: Frequency of self-reflection keywords in model responses. Higher noise assumptions correlate with increased self-reflection behavior across all keyword categories.

Results show that higher noise assumptions (larger $p$) consistently correlate with increased self-reflection behavior across all keyword categories. This suggests that noise-aware training facilitates the emergence of more sophisticated reasoning patterns by preventing premature convergence on superficial solution strategies.

## C.2 RESPONSE LENGTH ANALYSIS

Response length often correlates with reasoning depth (Muennighoff et al., 2025; Shen et al., 2025). Figure 8 compares average response lengths across models and baselines. S-GRPO consistently generates longer responses than baselines across all base models, with improvements ranging from 13% to 30%.

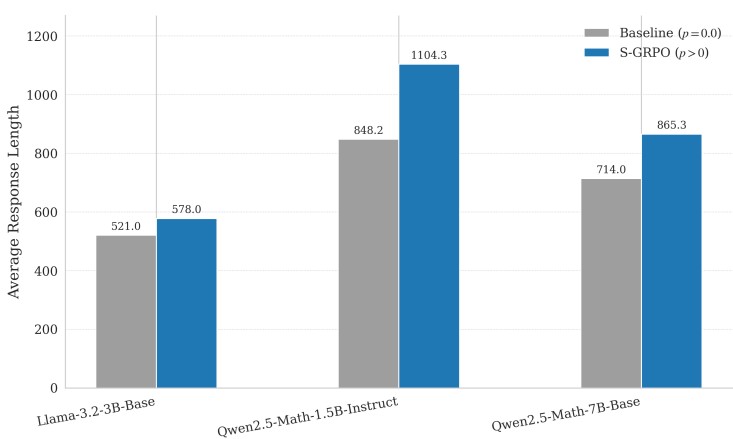

Figure 8: Comparison of response lengths for different models. S-GRPO consistently generates longer, more detailed responses across all base models.

# D    SUPPLEMENTARY EXPERIMENTAL RESULTS FOR THE NOISE HYPERPARAMETER $p$

## D.1    ABLATION STUDY OF OPTIMAL NOISE ASSUMPTIONS

Figure 9 examines the effect of different noise assumption levels $p$ on final performance. Due to computational constraints, we limit training to 300 steps and report the average of top-3 checkpoints.

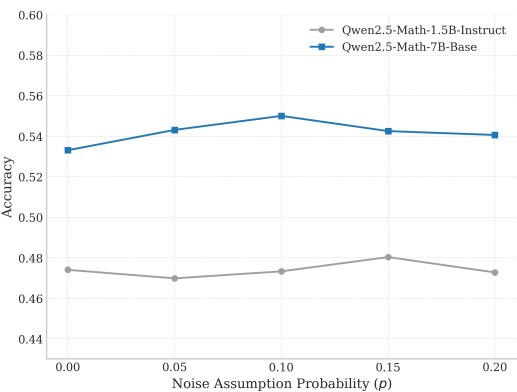

Figure 9: Effect of noise level $p$ on final performance. Optimal values differ by model scale, with larger models requiring lower noise assumptions.

Our findings reveal model-dependent optimal noise levels: Qwen2.5-Math-1.5B-Instruct achieves best performance at $p = 0.15$, while the larger Qwen2.5-Math-7B-Base performs optimally at $p = 0.10$. This pattern indicates that larger models exhibit lower optimal noise levels, suggesting they naturally produce fewer think-answer mismatches during training.

The higher optimal $p = 0.15$ for the 1.5B model has important implications: it excludes extreme groups ($k \in \{1, 7\}$ out of 8 rollouts) from training updates. This aggressive filtering provides net benefits for weaker models, where the risk of corrupted learning signals from highly imbalanced groups outweighs the potential information loss from excluding these samples.

## D.2    ROBUSTNESS TO SMALL NOISE PROBABILITIES $p$

In this section, we present the supplementary experimental results that show the robustness and effectiveness of the noise parameter $p$. The experiments were conducted on Qwen2.5-Math-1.5B-Instruct and Llama3.2-3B-Base, where we evaluate the performance of both GRPO and S-GRPO across different small values of $p$.

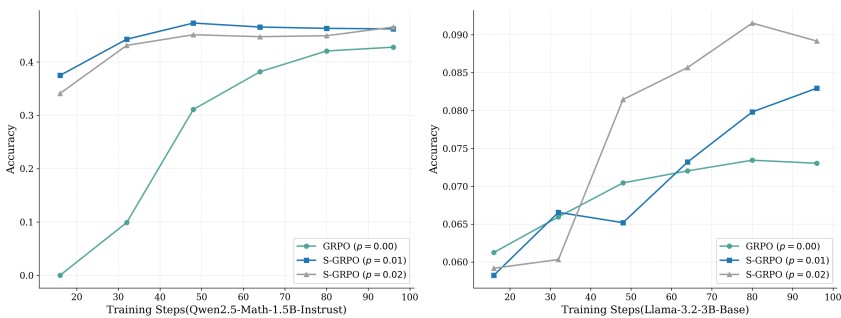

Figure 10: Accuracy comparison between GRPO and S-GRPO for Qwen2.5-Math-1.5B-Instruct (left) and Llama3.2-3B-Base (right) across different training steps. The results show that S-GRPO consistently outperforms GRPO with different values of $p \in \{0.01, 0.02\}$, indicating the robustness and effectiveness of the method across various models.

Table 3: Sensitivity of S-GRPO to the assumed noise rate $p$ and the number of training steps.

| Model | $p$ | Steps | Average |
|---|---|---|---|
| S-GRPO-1.5B$^\star$ | 0.10 | 256 | $49.7_{\pm0.3}$ |
| S-GRPO-3B$^\star$ | 0.10 | 480 | $9.8_{\pm0.4}$ |
| S-GRPO-7B$^\star$ | 0.15 | 496 | $56.0_{\pm0.4}$ |

Table 4: Pass@1 on MBPP and APPS under a small noise level $p$. S-GRPO consistently outperforms GRPO and the supervised baseline.

| Dataset | Method | Right count | Pass@1 |
|---|---|---|---|
| MBPP | Base Model | 94 / 195 | 0.482 |
| | GRPO | 96 / 195 | 0.492 |
| | S-GRPO | 104 / 195 | 0.533 |
| APPS | Base Model | 16 / 1000 | 0.016 |
| | GRPO | 21 / 1000 | 0.021 |
| | S-GRPO | 26 / 1000 | 0.026 |

As shown in Figure 10, S-GRPO consistently outperforms GRPO even with very small values of $p$, such as $0.01$. This suggests that the method remains effective when $p$ is set conservatively, and that the gains do not rely on fine-grained tuning. The improvements are especially noticeable for Qwen2.5-Math-1.5B-Instruct at shorter sequence lengths, where S-GRPO achieves substantially higher accuracy.

For the larger Llama-3.2-3B-Base, the improvements are more modest but still positive, and S-GRPO maintains an advantage over GRPO throughout training. These results indicate that S-GRPO can benefit models of different sizes and that small non-zero values of $p$ are sufficient in practice.

We then study how sensitive S-GRPO is to the assumed noise rate $p$ and the number of training steps. Table 3 summarizes the selected configurations. Across all three base models, the best results are obtained with noise rates ($p = 0.10$ or $0.15$) and training steps(256–496). In our sweeps, nearby values of $p$ and the number of steps produce similar Average scores, indicating that S-GRPO is not overly sensitive to precise tuning of these two hyperparameters as long as $p$ is chosen in a small range and the training budget is sufficient.

### D.3 Robustness on Code Generation Benchmarks

To further examine the robustness of S-GRPO beyond mathematical reasoning, we evaluate on two standard code generation benchmarks, MBPP and APPS. MBPP contains 974 short Python programming tasks that are solvable by entry-level programmers (Austin et al., 2021) and is widely used to assess function-level code synthesis. APPS is a large-scale benchmark built from coding competition problems with difficulty ranging from introductory to competition level (Hendrycks et al., 2021a).

Unlike the mathematical reasoning tasks in the main text, these benchmarks focus on code generation. They also allow us to test S-GRPO with different code models: Qwen2.5-Coder-1.5B-Instruct on MBPP and DeepSeek-Coder-1.3B-Instruct on APPS. This setting provides additional evidence that S-GRPO is robust across tasks, models, and domains.

For MBPP, we follow the standard problem formulation and randomly split 80% of the 974 problems for training and 20% for evaluation for computational efficiency. For APPS, we use the official training split and evaluate on a randomly sampled subset of 1,000 problems from the official test set, which covers a wide range of difficulty levels. For the RL methods, we use the same training hyperparameters as in the main text.

Table 4 shows that S-GRPO improves Pass@1 over GRPO by about 8% relatively on MBPP and about 24% on APPS, indicating that the robustness to small noise levels $p$ transfers from mathematical reasoning to code generation tasks.

Table 5: Performance comparison on AIME under the Avg@32 protocol. *Solved* denotes the number of correct solutions out of 960 sampled answers (30 problems × 32 samples), and *AIME Avg@32 (%)* is the corresponding average accuracy. Results marked with ⋆ indicate experiments we conducted under identical settings.

| Model | Solved | AIME Avg@32(%) |
|---|---|---|
| *Qwen2.5-Math-1.5B-Instruct* | | |
| Base Model | 93/960 | 9.67 |
| GRPO-1.5B⋆ | 97/960 | 10.10 |
| Dr. GRPO-1.5B⋆ | 94/960 | 9.79 |
| **S-GRPO-1.5B**⋆ | 98/960 | 10.20 |
| *Llama-3.2-3B-Base* | | |
| Base Model | 0/960 | 0 |
| GRPO-3B⋆ | 0/960 | 0 |
| Dr. GRPO-3B | 0/960 | 0 |
| **S-GRPO-3B**⋆ | 0/960 | 0 |
| *Qwen2.5-Math-7B-Base* | | |
| Base Model | 94/960 | 9.79 |
| GRPO-7B⋆ | 126/960 | 13.13 |
| Dr. GRPO-7B | 133/960 | 13.85 |
| **S-GRPO-7B**⋆ | 131/960 | 13.65 |

Table 6: Key Results of DAPO

| Method | Base Model | Benchmark / Dataset | Reported Result |
|---|---|---|---|
| DAPO | Qwen2.5-32B | AIME 2024 | 50 points (`avg@32`) |

## D.4 AIME RESULTS AND DISCUSSION

For completeness, we also evaluate our models on the AIME benchmark using the Avg@32 metric, as suggested in recent work. Table 5 reports the total number of correctly solved samples and the corresponding Avg@32 accuracy.

Table 5 shows that the absolute performance on AIME remains very low. All Llama-3.2-3B-Base variants obtain 0% Avg@32. The Qwen2.5-Math-1.5B and 7B models reach only around 10–14% Avg@32. S-GRPO matches or slightly improves on Dr. GRPO at both scales, but the gains correspond to only a few additional correct samples out of 960 and do not change the overall picture.

AIME problems are also much harder than the mathematical reasoning benchmarks used in the main text, and the base models we study have limited capability in this regime. When the Avg@32 accuracy is this low, all preference-optimization methods operate in a near-zero-reward regime, and the relative differences between methods are difficult to interpret.

For these reasons, we do not emphasize AIME in the main results and instead report it here in the appendix for completeness. We regard AIME as an interesting but extremely challenging benchmark for small and medium-sized models, and leave a more systematic study of this setting to future work.

## E RESULTS FROM DAPO AND GSPO

In our work, we focus on models that are smaller in size compared to DAPO and GSPO. DAPO uses a Qwen2.5-32B model and achieves 50 points (avg@32) on the AIME 2024 benchmark. This result surpasses the previous best of 47 points achieved by DeepSeek-R1-Zero-Qwen-32B, with DAPO using only approximately 50% of the training steps compared to the previous method. The dataset used for this task is the DAPO-Math-17K.

GSPO, which uses a Qwen3-30B model, has been applied to various benchmarks, including AIME'24, LiveCodeBench, and CodeForces. Although GSPO's specific numerical results are not publicly available, the authors claim that it outperforms GRPO in training efficiency and scaling.

While DAPO and GSPO use models around 32B in size, we focus on slightly smaller models that allow for better comparability, as the training data across these models is more consistent.

## F EXTENSION TO OTHER REWARD DOMAINS

Our main derivation in Sec. 3.2 assumes binary correctness rewards $r_i \in \{0, 1\}$ with symmetric label noise as in Eq. (5). In practice, other reward encodings such as $\{-1, 1\}$ or continuous scores may also be used. In this appendix, we clarify how S-GRPO extends to these settings.

### F.1 INVARIANCE UNDER REWARD RESCALING

Recall that the standardized advantages are defined as

$$a_i = \frac{r_i - \bar{r}}{\sigma_r}, \quad a_i^* = \frac{r_i^* - t}{\sigma_t},$$

and the optimal group weight is obtained by minimizing the mean-squared error in Eq. (8), leading to

$$w^\star = \mathrm{Cov}(a_i, a_i^*) = \frac{\mathrm{Cov}(r_i, r_i^*)}{\sigma_r \sigma_t} \qquad \text{(Eq. (10) revisited)}$$

as in Eq. (10). Under the symmetric flip model in Eq. (5), with latent true rewards $r_i^* \in \{0, 1\}$ of mean $t$ and observed rewards $r_i$ of mean $\bar{r} = k/N$, we have

$$\mathrm{Cov}(r_i, r_i^*) = (1 - 2p)\, t(1 - t),$$

which yields the closed-form expression $w^\star(N, k, p)$ in Eq. (11).

The key observation is that $w^\star$ can be written as

$$w^\star = (1 - 2p)\, \frac{\mathrm{Cov}(r_i, r_i^*)}{\sigma_r \sigma_t} = (1 - 2p)\, \mathrm{Corr}(r_i, r_i^*),$$

that is, the noise factor $(1 - 2p)$ times the correlation coefficient between observed and true rewards. The correlation term is invariant under any affine rescaling

$$\tilde{r}_i = a r_i + b, \quad \tilde{r}_i^* = a r_i^* + b, \quad a > 0,$$

because both the covariance and the product of standard deviations are scaled by the same factor $a^2$. Therefore, $w^\star$ is unchanged under any positive linear reparameterization of the reward domain.

As a concrete example, consider the common alternative encoding $\tilde{r}_i \in \{-1, 1\}$ obtained from $r_i \in \{0, 1\}$ via $\tilde{r}_i = 2r_i - 1$. If $t = \mathbb{E}[r_i^*]$ is the true success probability and $\mu = \mathbb{E}[\tilde{r}_i^*] = 2t - 1$, then

$$\mathrm{Var}(r_i^*) = t(1 - t), \qquad \mathrm{Var}(\tilde{r}_i^*) = 1 - \mu^2 = 1 - (2t - 1)^2 = 4t(1 - t).$$

Both $\mathrm{Cov}(\tilde{r}_i, \tilde{r}_i^*)$ and $\sigma_{\tilde{r}} \sigma_{\tilde{r}^*}$ are scaled by the same factor relative to the $\{0, 1\}$ case, so the resulting $w^\star$ is numerically identical under the two encodings. Thus S-GRPO applies to $\{-1, 1\}$ rewards, and more generally to any affine rescaling of binary rewards, without modification.

In pairwise preference settings, each comparison outcome (preferred vs. non-preferred) again induces a binary correctness indicator. By mapping the preferred item to 1 and the non-preferred item to 0 (or $\{-1, 1\}$), the same Bernoulli-based analysis applies to the induced labels and the resulting S-GRPO weighting.

### F.2 DISCUSSION OF CONTINUOUS REWARDS

The optimization problem in Eq. (8) itself does not rely on rewards being Bernoulli. For general scalar rewards $r_i \in \mathbb{R}$ and latent "clean" rewards $r_i^* \in \mathbb{R}$, we can still define standardized advantages

$$a_i = \frac{r_i - \bar{r}}{\sigma_r}, \quad a_i^* = \frac{r_i^* - t}{\sigma_{r^*}},$$

and consider the same objective

$$w^\star = \arg\min_w \mathbb{E}\big[(wa_i - a_i^*)^2\big], \qquad \text{(Eq. (8) revisited)}$$

where the expectation is taken over the joint distribution of $(r_i, r_i^*)$. Expanding the loss yields

$$w^\star = \mathrm{Cov}(a_i, a_i^*) = \frac{\mathrm{Cov}(r_i, r_i^*)}{\sigma_r \sigma_{r^*}},$$

in direct analogy to Eq. (10), regardless of the specific reward distribution.

In the Bernoulli case, we further specialize this expression using the symmetric flip model in Eq. (5) and the relationship between the observed mean $\bar{r} = k/N$ and the estimated true mean $t$ in Eq. (6), leading to the closed-form $w^\star(N, k, p)$ in Eq. (11). For more general continuous rewards, obtaining an explicit formula in terms of group-level statistics requires specifying a continuous noise model (for example, additive Gaussian noise around $r_i^*$) and computing $\mathrm{Cov}(r_i, r_i^*)$ and the variances under that model.

We do not pursue a full treatment of continuous-noise rewards in this work. Instead, we view Eq. (8) and the general solution $w^\star = \mathrm{Cov}(r_i, r_i^*)/(\sigma_r \sigma_{r^*})$ as a template that can be instantiated with alternative noise assumptions. Extending S-GRPO to process-level or continuous reward signals through such noise models is an interesting direction for future work.

## G    STATEMENT OF LLM USAGE

Large language models were used to aid or polish writing and to assist with retrieval and discovery of related work. All technical content, experimental design, and data analysis decisions were made independently by the authors, and the final manuscript was reviewed and edited by the authors.

