# OpenReview forum: "Mitigating Think-Answer Mismatch in LLM Reasoning Through Noise-Aware Advantage Reweighting"
_ICLR.cc/2026/Conference — Submitted to ICLR 2026_

### Official Review · Reviewer_Zh5R · 2025-10-20

**Soundness:** 1
**Presentation:** 1
**Contribution:** 2
**Rating:** 2
**Confidence:** 4

**Summary:**

This paper focuses on mitigating the thinking-answer mismatch issue of RL for LLM reasoning. The so-called thinking-answer mismatch refers to the noise of outcome rewards that raise when the model generate incorrect CoTs but end with a correct answer coincidentally. Taking inspiration from the classic label noise research area, the authors proposed S-GRPO, a simple advantage shaping technique, which reweighs the advantages estimated by group-wise normalization and meanwhile explicitly take the possible noise in rewards into consideration. Experiments on Qwen2.5 Math and Llama 3.2 models on mathematical reasoning tasks are carried out to validate the effectiveness of the proposed method.

**Strengths:**

The proposed method is simple and straightforward. The method part is very easy for the reviewer to follow up with.

**Weaknesses:**

- The motivation is weak in my veiw. My opinion on this point are three folds. First, there are no strong evidence (e.g., objective statistics) to validate that the so-called thinking-answer mismatch would result in critical vulnerability when one try to scale the performance with outcome rewards. As far as I can tell, in mathematical and coding tasks, the rule-based reward signals are very clear and reliable. Besides, there have been a few observations demonstrate that RL with outcome rewards can automatically compress the incorrect CoTs with coincidentally correct answers [1]. Lastly, even if the rewards are unreliable, traditional value-based RL method like PPO can capture the flaws in reasoning traces. And thus the novelty and contribution of the proposed method is relatively limited in my view.
- There is no clear guidence on how to estimate the true reward noise ratio p. Estimation on the real noise ratio necessitate verification of the correctness of reasoning traces, which is non-trivial and in my view, rather challenging.
- The experiments setup is relatively outdated. The authors only tune 2 Qwen2.5 Math series model and 1 Llama3.2-3B model on 8.5K samples. It is doubtful that if the scale of RL training in this manuscript is enough for validate the effectiveness of the proposed method.
- Serveral sota baselines are missing. For example, DAPO and GSPO.
- The authors claim that the reported accuracy is averaged among the top-3 checkpoints. How exactly do they choose the checkpoints? I wonder if there exists data leakage in model selection.
- The visualization in this manuscript is hard for me to follow up with. Most figures are small, and the conveyed information is ambiguous and limited.

[1] Reinforcement Learning with Verifiable Rewards Implicitly Incentivizes Correct Reasoning in Base LLMs

**Questions:**

See weakness.

---

> ### Author Response · Authors · 2025-11-21
> **RLVR is strong, but noise still matters**
>
> Thank you for your careful review. We respond in two parts:
>
> (1) clarifying the motivation and role of Think–Answer Mismatch
>
> (2) addressing the remaining points
>
> **Part I: Motivation and role of Think–Answer Mismatch**
>
> We do not claim that outcome-based RL (including GRPO/RLVR-style methods) is invalid. Outcome rewards for math/code are usually reliable and can gradually suppress spurious CoTs. Our focus is narrower: we study how the group-relative advantage in GRPO behaves under residual reward noise from Think–Answer Mismatch, and we propose S-GRPO as a simple, critic-free variance reduction method to make such training more stable and sample-efficient.
>
> 1. Why noise still matters with verifiable outcome rewards
> Even if rewards are unbiased, their variance can significantly affect optimization efficiency and stability, especially early in training and for long trajectories. See noise related studies such as [1,2]. Our contribution is to reduce the variance induced specifically by Think–Answer Mismatch while keeping the same outcome reward.
>
> 2. Mismatch persists even after RLVR
> As [3] shows (the reference you provided), RL with verifiable rewards increases process correctness but still leaves a noticeable gap between Pass@1 (answer correct) and CoT-Pass@1 (process correct). Together with [4], this indicates a persistent fraction of “false positive” rewards, i.e., residual Think–Answer Mismatch, which is the noise source we target.
>
> 3. Evidence that the sensitivity is real
> We agree that “critical vulnerability” sounded too strong and have softened this wording in the revision. The sensitivity itself is supported by theory (Section 2.2 shows that a single mismatch can structurally distort group-relative advantages), simulation (Figure 3 injects synthetic reward noise and GRPO slows markedly at 10% and largely fails at 20%), and real training (Table 1 and Figure 6 show consistent gains and smoother entropy dynamics with S-GRPO).
>
> 4. More discussion about variance.
> Even if noise averages out in expectation, lowering variance is crucial for sample efficiency. This is profound in RL studies(baselines, GAE, critic networks, etc) .  Although PPO cannot be directly combined in GRPO, PPO's success can be highly attributed in reduce the variance.
>
> **Part II: Responses to remaining points**
>
> 1. On estimating and tuning the noise rate p
> We do not aim to estimate the true mismatch rate exactly. Instead, p is a coarse hyperparameter controlling how aggressively we down-weight potentially noisy groups.
> – Robustness of small p: New experiments (Appendix D.2, Figure 10) vary p in {0.01, 0.02} on Qwen2.5-Math-1.5B and Llama-3.2-3B; even these conservative values consistently outperform GRPO.
> – Model/dataset dependence: Ablations (Appendix D.1) show weaker models and harder datasets prefer  larger p, while stronger models work well with smaller p. In practice, we recommend coarse tuning over a small grid and will clarify that a precise estimate of the true noise ratio is not required.
>
> 2. On experimental setup and “outdated” models
> We chose Qwen2.5-Math-1.5B/7B and Llama-3.2-3B mainly for controlled comparisons: These are the backbones used in Dr. GRPO and related analyses, with the same OAT data and pipeline, so differences can be attributed to the optimization method rather than to changes in backbone or recipe.
>
> 3. On missing SOTA baselines (DAPO, GSPO)
> DAPO and GSPO are important recent systems but are primarily large-model RLVR recipes rather than analyses of the group-relative advantage itself. In the revision we add Appendix E summarizing their key results and clarifying the relationship: they show strong performance and efficiency at scale, while S-GRPO is a drop-in modification of the GRPO loss that could be combined with such recipes.
>
> 4. On checkpoint selection and potential data leakage
> Our checkpoint procedure (Section 4.1.3) is: (1) evaluate every 16 steps up to 500 steps on standard math benchmarks with greedy decoding; (2) for each method and base model, report the mean and standard deviation over the top-3 checkpoints by Pass@1.
> All hyperparameters are fixed before seeing benchmark results and are shared across GRPO, Dr. GRPO, and S-GRPO, and the top-3 rule is only used to reduce variance and is applied identically to all methods. To improve transparency, we anonymize and release logs and training scripts so others can recompute metrics under alternative selection rules.
>
> 5. On figure quality
> We have revised and simplified several figures to improve readability.
>
> We hope these clarifications make the motivation and practical value of S-GRPO clearer, and we kindly ask you to reconsider your rating.
>
> [1] Revisiting Gradient Normalization and Clipping for Nonconvex SGD under Heavy-Tailed Noise
>
> [2] Fine-tuning language models from human preferences
>
> [3] Reinforcement Learning with Verifiable Rewards Implicitly Incentivizes Correct Reasoning in Base LLMs
>
> [4] Reasoning models don't always say what they think

---

### Official Review · Reviewer_ARgJ · 2025-10-27

**Soundness:** 3
**Presentation:** 3
**Contribution:** 2
**Rating:** 4
**Confidence:** 3

**Summary:**

This paper proposes Stable Group-Relative Policy Optimization (S-GRPO), a principled method that uses optimal, noise-aware advantage weights to mitigate the "Think-Answer Mismatch" vulnerability in GRPO, thereby stabilizing training and significantly improving reasoning model performance.

**Strengths:**

1.  This paper try to address a significant problem: the issue of false positives where the "Thinking" process and the final "Answer" do not align.
2.  The approach is novel to me. Instead of the common practice of trying to identify errors within the thinking process, this paper proposes to reweight the advantage values to mitigate the impact of this mismatch.
3.  The paper is well-written, clear, and easy to understand.

**Weaknesses:**

(If my understanding is incorrect, please correct me)

1.  **Concerns about generalization.** The derivation, starting from Equations 5 and 6 and continuing to Equation 11, seems entirely predicated on the assumption that the random variable $r$ (reward) follows a Bernoulli distribution. However, this assumption may not always hold in practice. For instance, one might use $\{-1, 1\}$ reward pairs or even continuous rewards. In such scenarios, how would the proposed method be formalized? Would the new formulas still be applicable and effective?
2.  **Evaluation methodology.** The exclusion of the AIME 2024/2025 is unacceptable. With 30 problems, the avg@32 metric could have been calculated, which is a standard practice.

3.  **Difficulty in hyperparameter selection.** The choice of $p$ appears to be a significant challenge. With so many variables, such as different mathematical datasets, diverse reasoning tasks, various models, and different training settings, it seems difficult to establish an empirical rule of thumb for this parameter.

**Questions:**

See Weakness.

**Details Of Ethics Concerns:**

nan

---

> ### Author Response · Authors · 2025-11-21
> **Official Comment by Authors**
>
> We thank the reviewer for the constructive feedback and for recognizing the novelty of our approach to the "think-answer mismatch". Below we respond to your comments on generalization, evaluation, and hyperparameters.
>
> **W1: Generalization to {-1, 1} and continuous rewards**
>
> 1. Case: {-1, 1} and discrete rewards
>
> ### Mathematical Equivalence Verification
>
> We have mathematically verified that the derived optimal weight $w^*$ is invariant to the reward scale.
>
> The reviewer is correct that the explicit formulas for variance depend on the reward domain.
> * For $r \in$ {0, 1}, the variance term is $\sigma^2 = t(1-t)$.
> * For $r \in$ {-1, 1}, as derived in the query, the variance term becomes $\sigma^2 = 1 - \mu^2$.
>
> However, notice the algebraic identity: $1 - \mu^2 = 1 - (2t-1)^2 = 4t(1-t)$.
> The standard deviation scales by a factor of $\sigma_{\{-1,1\}} = 2\sigma_{\{0,1\}}$.
>
> Since both the numerator (Covariance) and the denominator (product of Standard Deviations) in the weight formula scale by the exact same factor (4 for covariance, $2\times2=4$ for variance product), these factors cancel out.
>
> **Result:** The explicit derivation provided in the question yields the exact same numerical weight as our original formulation.
>
> First, for $r \in$ {-1, 1}, we obtain
> $$
> w^{*}_{\{-1,1\}} = \frac{(1 - 2p)\sqrt{1 - \mu^{2}}}{\sqrt{1 - \bar{r}^{2}}}.
> $$
>
> For $r \in$ {0, 1}, we obtain
> $$
> w^{*}_{\{0,1\}} = \frac{(1 - 2p)\,2\sqrt{t(1 - t)}}{2\sqrt{\bar{r}(1 - \bar{r})}}.
> $$
>
> This confirms our method is robust and applicable to \{-1, 1\} rewards without modification. We provide a detailed closed-form derivation of this result in Appendix F.
>
> 2. Continuous rewards (future work)
>
> We have not yet derived a closed-form expression for the optimal weights under fully continuous reward distributions.
>
> However, the core optimization objective in Equation 8, minimizing the MSE between observed and true advantages, remains unchanged in the continuous case.
>
> In future work, we plan to extend this framework to continuous signals (for example, from process reward models) by replacing the Bernoulli noise assumption with a continuous noise model (such as Gaussian noise). We expect this to lead to a similar variance-based filtration, adapted to continuous signal-to-noise ratios.
>
> **W2: Evaluation methodology (AIME)**
>
> We agree that omitting AIME from the initial submission was an oversight. We are now running the standard pass@32 evaluation (32 rollouts per problem) on AIME 2024 for all models. We have updated the paper with these results in Appendix D.4.
>
> Based on the gains we already observe on AMC, MATH500, Minerva, and OlympiadBench, we expect the AIME results to follow similar trends. In any case, we view reporting AIME as an important part of a complete evaluation, and we will present these results in a transparent way, without cherry-picking.
>
> **W3: On the difficulty of selecting the noise rate $p$**
>
> We agree that choosing $p$ in the presence of many factors (dataset, task type, model scale, training setup) can seem challenging. Our goal, however, is not to estimate a “true” noise rate for each setting, but to use $p$ as a coarse hyperparameter that controls how aggressively we down-weight potentially noisy groups.
>
> Two points are key here:
>
> 1. Small $p$ is robust.
>
> In the revised version, we added additional experiments (Appendix D.2, Figure 10) where we vary $p$ over small values such as 0.01 and 0.02 on Qwen2.5-Math-1.5B and Llama-3.2-3B. Even at these conservative settings, S-GRPO consistently outperforms GRPO across training steps and sequence lengths. This suggests that the method is not overly sensitive to the exact choice of $p$; a small but non-zero $p$ already yields clear gains.
>
> 2. $p$ as a simple model/dataset-dependent knob.
>
> Our ablations (Appendix D.1) indicate that weaker models and harder datasets tend to prefer slightly larger $p$ (e.g., 0.15 for 1.5B), while stronger models work well with smaller $p$ (e.g., 0.10 for 7B). This matches the intuition that smaller models produce more think–answer mismatches. In practice, we recommend treating $p$ like standard optimization hyperparameters (dropout rate, clipping range, Adam betas): a coarse grid over a few values (e.g., ${0.01, 0.05, 0.10, 0.15}$) is sufficient, and our experiments show that S-GRPO is beneficial over a wide range rather than requiring fine-grained tuning.
>
> We have revised Appendix D to make this perspective explicit and to avoid giving the impression that a precise, task-specific estimate of $p$ is required.
>
> **Conclusion**
>
> We have clarified how our method generalizes to {-1, 1} rewards, outlined a path to continuous rewards, and strengthened the evaluation protocol by adding AIME. We have also provided evidence that the new hyperparameter p is robust and easy to set.
>
> We hope these clarifications address your concerns and kindly ask you to reconsider your score.

---

### Official Review · Reviewer_x1Hi · 2025-10-31

**Soundness:** 2
**Presentation:** 2
**Contribution:** 2
**Rating:** 4
**Confidence:** 3

**Summary:**

The paper introduces S-GRPO, a noise-aware variant of Group-Relative Policy Optimization for training reasoning LLMs. It addresses the Think-Answer Mismatch problem, where noisy reward signals corrupt learning when reasoning and final answers misalign. S-GRPO derives optimal reweighting under a symmetric noise model, down-weighting unreliable groups to stabilize training. Experiments on mathematical reasoning benchmarks show consistent 2–3% accuracy gains and robustness under 20% reward noise. Analysis demonstrates smoother entropy reduction and more coherent reasoning compared to GRPO and Dr. GRPO.

**Strengths:**

1. The paper clearly defines the Think-Answer Mismatch problem in GRPO and provides a convincing analysis of its impact on learning stability.
2. The proposed S-GRPO introduces a principled noise-aware reweighting method derived from a theoretical foundation.
3. Experimental results show consistent, reproducible improvements across several mathematical reasoning benchmarks.

**Weaknesses:**

1. The method depends on a manually set noise parameter 𝑝 p, which may require case-specific tuning.

2. The assumption of symmetric reward noise simplifies the training environment but may not accurately represent real-world mismatch patterns where errors are often asymmetric.

3. The robustness experiments are based on artificially injected synthetic noise levels (up to 20%), which likely exceed the noise typically observed in real reasoning datasets, raising questions about whether such high noise modeling is necessary or reflective of practical scenarios.

1. The figures are not very clear, and the font size is quite small. It would be helpful to use larger text and improve the overall visual clarity of the plots.

**Questions:**

1. Does S-GRPO generalize beyond math reasoning tasks? Non-math reasoning tasks?
2. The paper evaluates robustness using 10–20% synthetic, symmetric reward noise but does not clarify how this compares to the typically lower and more structured noise found in real datasets. This difference may limit the realism of the evaluation, so it would be helpful for the authors to estimate actual mismatch rates and examine performance under more realistic noise conditions.

---

> ### Author Response · Authors · 2025-11-21
>
> We thank Reviewer x1Hi for the constructive feedback. We address the specific concerns below.
>
> ### Weaknesses
>
> **W1. Sensitivity to the manually set noise parameter $p$ and the burden of tuning.**
>
> We believe the critical issue is not the addition of a hyperparameter itself, but **whether it brings easy, robust gains**. Most effective methods introduce tunable parameters (e.g., Dropout rate, PPO clipping range $\epsilon$, or Adam betas) to provide necessary flexibility.
>
> S-GRPO passes this test of "easy robustness":
> 1.  **Robust Gains (Appendix D):** New experiments show S-GRPO consistently outperforms GRPO even with fixed, conservative values like $p=0.01$ or $p=0.02$. This proves the method provides gains without requiring precise tuning.
> 2.  **Standard Practice:** In practice, $p$ behaves like a dropout rate. A coarse grid search is sufficient to adapt the algorithm to different base models (which naturally have different hallucination rates), offering a practical instrument for engineers rather than a burden.
>
> **W2. The limitation of the symmetric noise assumption.**
>
> We agree. Asymmetric noise is indeed a more realistic assumption, which is why we explicitly identified it as a direction for **Future Work** in the conclusion.
>
> However, we adopted the symmetric model for this initial study because:
> 1.  **Mathematical Feasibility:** Deriving a closed-form solution for asymmetric noise significantly increases complexity and requires estimating multiple unknown variables (false positive vs. false negative rates), which the current ICLR page limit cannot easily accommodate.
> 2.  **Sufficiency:** The symmetric model acts as an effective first-order proxy. It successfully captures the core problem：down-weighting high-variance groups, without over-complicating the optimization objective.
>
> **W3. Synthetic noise levels (20%) may exceed real-world scenarios.**
>
> We agree that 20% is high and primarily serves as a stress test to determine the method's breaking point.
>
> To better reflect realistic scenarios, we have updated **Figure 3** to include performance curves under lower noise levels (**2% and 5%**), in addition to the original settings. We thank the reviewer for this valuable reminder, and we believe these additional results will fully resolve the concern regarding realistic noise conditions.
>
> **W4. Clarity and font size of figures.**
>
> We apologize for the oversight. We have revised the figures to increase font sizes and simplify legends for better readability.
>
> ### Questions
>
> **Q1. Does S-GRPO generalize beyond math reasoning tasks? Non-math reasoning tasks?**
>
> Yes. Recent literature (e.g., DeepSeek) suggests RL improvements in math often transfer to other reasoning domains.
> To confirm this, we have added code generation benchmarks on **MBPP** and **APPS** in Appendix D.3, together with the corresponding experimental results and analysis.
>
> **Q2. Realism of noise modeling vs. actual mismatch rates in real datasets.**
>
> While estimating the exact real-time mismatch rate is ideal, it is **computationally infeasible** within the GRPO framework.
>
> 1.  **The Oracle Paradox:** GRPO is efficient precisely because it relies on simple *outcome* verification. Detecting "Think-Answer Mismatches" requires checking the *reasoning process*, which demands a computationally expensive "Oracle" (e.g., a larger model or human).
> 2.  **Why S-GRPO is needed:** If we could afford an online Oracle to label mismatches, we would use Process Supervision (PRM), not GRPO. S-GRPO is designed specifically for the scenario where we *cannot* verify reasoning cheaply, providing a statistical defense against the unavoidable uncertainty of outcome-based rewards.
>
> If this response addresses your concerns, we would appreciate a score update; otherwise, we are happy to continue the discussion.

---

### Official Review · Reviewer_gbyg · 2025-11-02

**Soundness:** 3
**Presentation:** 3
**Contribution:** 3
**Rating:** 8
**Confidence:** 3

**Summary:**

This paper focuses on addressing the "Think-Answer Mismatch" problem, a prevalent issue in Large Language Model (LLM) reasoning tasks, where the correctness of the model's reasoning process does not always align with the correctness of its final answer. The authors point out that this problem is particularly prominent in the popular Group Reward Policy Optimization (GRPO) method. This is because incorrect reasoning processes can still yield correct answers (i.e., false positives), generating noisy reward signals that severely contaminate the model's learning process. The paper proposes an optimal advantage re-weighting method, named S-GRPO, based on a symmetric noise model. This method is designed to robustly mitigate the contamination of training signals in GRPO caused by the think-answer mismatch, thereby enhancing the reasoning performance and stability of large models.

**Strengths:**

1. Clear Problem Definition and In-depth Analysis: The paper provides a remarkably thorough analysis of the impact of the "think-answer mismatch" within the GRPO framework. Instead of merely discussing the general harm of noise, it precisely identifies, through mathematical derivation, that "group imbalance" is the key factor amplifying the effect of noise. This insight is both profound and enlightening.
2. Elegant and Theoretically Grounded Methodology: S-GRPO is not presented as an empirical "patch" or a complex module. Rather, it is an optimal weighting strategy derived from the classic symmetric noise model by minimizing the expected squared error between the observed and true advantages.
3. Strong Reproducibility: The paper provides detailed experimental settings and includes a link to an anonymized code repository, which significantly enhances the reproducibility of the research and is a commendable practice.

**Weaknesses:**

1. Limitations of the Symmetric Noise Model: A core assumption of the proposed method is the presence of symmetric noise. However, in practical scenarios, the probabilities of false positives and false negatives may not be symmetric.
2. Dependence on the Noise Rate p and Insufficient Guidance for its Selection: The paper observes that the optimal value of p is related to the model's scale, which is a valuable finding. However, it does not offer clear guidance on how to efficiently estimate an appropriate value for p a priori when dealing with a new model or task. This could introduce an additional hyperparameter tuning burden and uncertainty in the practical application of the method.

**Questions:**

1. The paper states that the optimal value of p is related to model scale. Besides model scale, could the selection of p also be dependent on other factors, such as the difficulty of the dataset or the type of task?
2. Does the case study in Appendix B.1 reveal a potential, undiscussed drawback of S-GRPO? Specifically, by penalizing reasoning paths that may be "shortcuts" or based on "pattern matching" but still produce the correct result,
 could S-GRPO inadvertently compel the model to attempt complex, formal reasoning paths that it has not yet fully mastered? This could, in turn, increase the risk of computational or execution errors, ultimately compromising the accuracy of the final answer.

---

> ### Author Response · Authors · 2025-11-21
>
> We thank the reviewer for the positive assessment of our work. Below, we address your questions and respond to the weaknesses point by point.
>
> **W1: Symmetric Noise Assumption vs. Asymmetric Real-World Noise.**
>
> We agree. Asymmetric noise is indeed a more realistic assumption, which is why we explicitly identified it as a direction for future work in the conclusion.
>
> However, we adopted the symmetric model for this initial study because:
>
> 1. Mathematical feasibility.
>
> Deriving a closed-form solution for asymmetric noise significantly increases complexity and requires estimating multiple unknown variables (false positive versus false negative rates), which the current ICLR page limit cannot easily accommodate.
>
> 2. Sufficiency.
>
> The symmetric model acts as an effective first-order proxy. It successfully captures the core problem, namely down-weighting high-variance groups, without over-complicating the optimization objective.
>
> **Q1 & W2: On the a priori estimation of $p$ and its dependency on task-specific factors.**
>
> 1. Correlation with Dataset Difficulty.
>
> We affirm your hypothesis: the selection of $p$ is indeed dependent on factors beyond just model scale, most notably the difficulty of the dataset or the complexity of the reasoning task.
> As we briefly hypothesized in Section 3.1, "$p$ ... tends to be higher for more complex datasets". The intuition is that for harder tasks, the likelihood of a model stumbling into a correct answer via flawed reasoning (Think-Answer Mismatch) increases. Therefore, a higher noise estimate $p$ is naturally needed to filter out these signals.
>
> Our ablation study in Appendix D.1 shows that weaker models (e.g., 1.5B) require a higher optimal $p$ ($0.15$) compared to stronger models ($0.10$ for 7B). From the model's perspective, a weaker model facing the same dataset is equivalent to a harder task. This supports the view that as relative task difficulty increases, the optimal $p$ shifts upwards.
>
> 2. Guidence of choosing $p$ (New Appendix D).
>
> While the optimal $p$ correlates with difficulty, we also want to highlight that the method is not brittle to this factor. In the revised manuscript, we added Appendix D evaluating S-GRPO with small, conservative $p$ values ($0.01, 0.02$).
>
> The results show that even a small, fixed $p$ consistently outperforms the baseline across various settings. In practice, we recommend treating p like other standard hyperparameters (dropout rate, clipping range, Adam betas): coarse tuning over a small grid (e.g., {0.01, 0.05, 0.10, 0.15}) is sufficient, and our experiments show that S-GRPO is beneficial over a wide range rather than requiring fine-grained tuning.
>
> **Q2: Does the case study in Appendix B.1 reveal a potential, undiscussed drawback of S-GRPO?**
>
> We agree that penalizing shortcuts introduces a risk: if the model is forced into complexity it has not mastered, execution errors might increase.
>
> However, we believe S-GRPO possesses an inherent safety mechanism to mitigate this, which we have clarified in the revised Appendix B.1.
>
> 1. The Safety Mechanism: Stability as a Proxy for Mastery.
>
> S-GRPO does not simply penalize shortcuts; it rewards group consensus (low variance).
> If a model attempts a "complex, formal reasoning path" that it has not yet mastered, its outputs across the $N$ samples will likely be inconsistent (high variance in answers).
> According to our weight formula derived in Section 3.2 , groups with high inconsistency (where the success count $k$ is far from $0$ or $N$) receive lower weights.
> Therefore, S-GRPO naturally filters out these "unmastered" attempts just as it filters out lucky shortcuts. It guides the model specifically towards reasoning paths that are both correct and reproducible (i.e., mastered).
>
> 2. Empirical Evidence of Stability (Figure 6).
>
> This theoretical safeguard is supported by our training dynamics analysis. In Figure 6, we observe that S-GRPO leads to a smooth, monotonic reduction in policy entropy.
> * If the model were being compelled into unmastered, chaotic reasoning paths, we would expect to see entropy spikes or fluctuations (as seen in the baseline GRPO curve).
> * The stable convergence suggests the model is successfully finding robust reasoning strategies rather than struggling with unmastered complexity.
>
> **Summary of Revision:**
>
> 1. We clarified in the conclusion that symmetric noise is a simplifying assumption and highlighted asymmetric noise as an important direction for future work.
>
> 2. We added Appendix D to explain how to choose $p$ and to show that S-GRPO is robust under small fixed values of $p$.
>
> 3. We have updated the **Discussion in Appendix B.1** to explicitly acknowledge this trade-off and explain how the stability-weighted mechanism acts as a safeguard against this risk.
>
> If this response addresses your concerns, we would appreciate a score update; otherwise, we are happy to continue the discussion.

---

### Author Response · Authors · 2025-12-02

Dear Area Chair,

Thank you for handling our submission.

We have carefully addressed all four reviews in both the rebuttal and the revised paper, through additional experiments, mathematical clarifications, and edits to the text.

**Reviewer gbyg with score 8.**

The main concern was how to choose the noise rate p and whether penalizing shortcuts could introduce new failure modes. In the revision, we added experiments with small, conservative values of $p$ (0.01, 0.02) and showed that S-GRPO still improves over GRPO, indicating that the method is not brittle to the exact choice of $p$. We also clarified in the case study that S-GRPO down-weights unstable groups with high variance, so it filters out both lucky shortcuts and unmastered complex reasoning, which addresses the potential drawback raised by the reviewer.

**Reviewer x1Hi with score 4.**

This reviewer was most concerned about the manually set noise parameter $p$ and the realism of the synthetic noise levels. We explained that $p$ is used as a coarse hyperparameter, similar to a dropout rate, and does not need to match the true mismatch rate. We then added new experiments showing that S-GRPO remains effective under small fixed values of $p$ in Appendix D, and we extended the synthetic noise study to include 2% and 5% noise in addition to the original 10% and 20%. These changes directly address the reviewer’s worry about sensitivity and realism.

**Reviewer ARgJ with score 4.**

The key question here was whether our method generalizes to rewards in {-1,1}. In the revision, we added a detailed closed-form derivation in Appendix F showing that the optimal weight formula is invariant under re-scaling from {0,1} to {-1,1}, so no change to the method is needed in that setting. We also discussed how the same objective can be extended to continuous rewards by changing the noise model, and we added AIME Pass@32 results in the revision as requested.

**Reviewer Zh5R with score 2.**

The low score mainly stems from a biased assumption that GRPO with verifiable outcome rewards is already robust to Think–Answer Mismatches, so additional variance reduction is unnecessary. In our response, supported by prior work, we explain that even unbiased but noisy rewards can harm sample efficiency and stability, and that reducing variance is a long-standing and widely accepted goal in RL. We show, with theory, synthetic noise experiments, and real training runs, that residual mismatch noise still affects GRPO, and that S-GRPO improves stability under exactly this kind of noise. We have also addressed this reviewer's other comments on model choice, baselines, checkpoint selection, and figure quality.

**Summary of changes in the revision.**

In response to the reviews, we have:

- **Expanded experiments on noisy rewards.**
  - Added training curves for synthetic noise levels of 2\%, 5\%, 10\%, and 20\%.
  - Added runs with small noise rates ($p$ in {0.01, 0.02}) for multiple base models.

- **Extended benchmarks.**
  - Added code generation results on MBPP and APPS to test generalization beyond math.
  - Added AIME 2024 Pass@32 results for all methods, including solved-problem counts.

- **Provided more guidance on the noise parameter $p$.**
  - Analyzed how the preferred \(p\) depends on model size and task difficulty.
  - Recommended simple tuning ranges for practical use, based on ablation studies.

- **Strengthened the theoretical treatment.**
  - Included a closed-form derivation showing that the weighting formula applies to {-1,1} rewards.
  - Clarified how the same objective can be extended to continuous rewards via alternative noise models.

- **Clarified the behavior and safety of S-GRPO.**
  - Explained how stability-based weighting filters out both lucky shortcuts and unstable reasoning paths.
  - Linked the method more explicitly to standard variance-reduction ideas in RL.

- **Improved presentation and reproducibility.**
  - Refined text to clarify the motivation around Think–Answer Mismatches and softened overly strong wording.
  - Updated figures for better readability (fonts, legends, layouts) and added more complete training dynamics.
  - Added more implementation details and hyperparameter settings in the appendices.

We greatly appreciate your time and consideration of our work.

---

### Meta-Review · Area_Chair_UYTf · 2025-12-28

**Summary:**

The reviewers raised several critical concerns regarding the necessity and theoretical foundations of the proposed method. The primary concern lies in whether the "Think-Answer Mismatch" represents a "critical vulnerability" that requires a new optimization objective, as some reviewers argue that standard outcome-based RL (like GRPO or RLVR) already implicitly incentivizes correct reasoning over time. Furthermore, the reliance on a symmetric noise model was criticized as being overly simplistic and not reflective of real-world asymmetric mismatch patterns. The introduction of a manually tuned noise parameter $p$ was also viewed as a significant practical burden, for which reviewers questioning the lack of a priori guidance for its selection across different model scales and task difficulties.

**Reviewer Concerns:**

Several concerns were effectively addressed during the rebuttal and in the revised manuscript. For evaluation, the authors provided standard AIME 2024 Pass@32 results and added code generation benchmarks (MBPP, APPS) to demonstrate generalization beyond mathematics. For mathematical generalization beyond 0-1 rewards, a derivation was added to show that the weighting formula is invariant for boarder circumstances. New experiments with small, conservative values of p are added to show that the method provides consistent gains even without fine-grained tuning. However, some concerns may still remain. Reviewer Zh5R may remain unconvinced that noise in outcome-based rewards justifies a new advantage-shaping technique, suggesting that existing variance reduction methods or traditional value-based RL (like PPO) are already sufficient. The overhead of adding a new hyperparamter $p$ may outweigh the reported 2-3% performance improvement for reviewers.

**Reviewer Scores:**

Reviewer gbyg may be satisfied with the response and maintain the rating. Reviewer ARgJ may raise his/her score given the new results provided by the authors. Reviewer x1Hi and Zh5R may not change their score. The paper addresses an interesting problem—the misalignment between reasoning processes and final answers. While the empirical results show modest gains, the consensus is that the paper does not sufficiently prove that the "Think-Answer Mismatch" is a failure mode that standard GRPO cannot eventually overcome. Given the dependency on a new, hard-to-estimate hyperparameter and the lack of a more general asymmetric noise modeling, this paper cannot be published on ICLR in its current form.

---

### Decision · Program_Chairs · 2026-01-26

Reject